# LC-MS Analysis of Serum for the Metabolomic Investigation of the Effects of Pulchinenoside b4 Administration in Monosodium Urate Crystal-Induced Gouty Arthritis Rat Model

**DOI:** 10.3390/molecules24173161

**Published:** 2019-08-30

**Authors:** Shang Lyu, Ruowen Ding, Peng Liu, Hui OuYang, Yulin Feng, Yi Rao, Shilin Yang

**Affiliations:** 1National Pharmaceutical Engineering Center for Solid Preparation in Chinese Herbal Medicine, Jiangxi University of Traditional Chinese Medicine, Nanchang 330006, China; 2State Key Laboratory of Innovative Drug and Efficient Energy-Saving Pharmaceutical Equipment, Nanchang 330006, China

**Keywords:** pulchinenoside b4, gouty arthritis, monosodium urate crystal, UPLC-QTOF-MS/MS, metabolomics, pharmacodynamics, pathway, potential biomarkers, multivariate analysis

## Abstract

Gouty arthritis (GA) is commonly caused by deposition of monosodium urate (MSU) crystals within the joint capsule, bursa, cartilage, bone, or other periarticular tissues after chronic hyperuricemia. Clinically, GA is characterized by acute episodes of joint inflammation, which is most frequently encountered in the major joints, and also has a significant impact on quality of life. Pulchinenoside b4(P-b4) has a wide range of biological activities, including antitumor, anti-inflammatory, antiviral and immunomodulatory activities. Currently, the anti-GA activity and metabolomic profiles after being treated by P-b4 have not been reported. In this paper, for the first time, we have performed a non-targeted metabolomics analysis of serum obtained from an MSU crystal-induced GA rat model intervened by P-b4, using ultra-performance liquid chromatography coupled to quadrupole time-of-flight tandem mass spectrometry. In this study, the main pharmacodynamics of different dosing methods and dosages of P-b4 was firstly investigated. Results have shown that P-b4 possesses high anti-inflammatory activity. These results demonstrated changes in serum metabolites with 32 potential biomarkers. Arachidonic acid, sphingolipid, and glycerophospholipid metabolism are considered to be the most relevant metabolic pathway with P-b4 treatment effect in this study. Moreover, the changes of metabolites and the self-extinction of model effects within 24 h reveals important information for GA diagnostic criteria: The regression of clinical symptoms or the decline of some biochemical indicators cannot be regarded as the end point of GA treatment. Furthermore, our research group plans to conduct further metabolomics research on the clinical course of GA.

## 1. Introduction

Gouty arthritis (GA) is commonly caused by deposition of monosodium urate (MSU) crystals within the joint capsule, bursa, cartilage, bone, or other periarticular tissues after chronic hyperuricaemia (serum urate levels >7 mg/L or 420 μmol/L). The abridged general view of GA is provided in Figure 1. According to the statistics, its morbidity and prevalence are increasing year by year worldwide (about 1–2%, 2018), especially in developing countries [1,2,3]. Clinically, GA is characterized by acute episodes of joint inflammation, which are most frequently encountered in the major joints, especially in the first metatarsophalangeal joint, ankle, and foot joints. Other complications include chronic renal injury, ureteral calculi, tophi (formation of urate deposits), and arthritis malformation [4,5]. Typically, the clinical course of GA includes 3 periods: acute arthritis, intermission, and chronic arthritis. Clinical manifestations, laboratory tests, and X-ray examinations are helpful for the diagnosis; however, a complete diagnosis should be made by examining the synovium or synovial fluid to detect MSU crystal, on account of the serum urate of psoriatic arthritis and rheumatoid arthritis is also increased occasionally. 

Pulchinenoside b4(P-b4) belongs to pentacyclic triterpenoid saponins (lupane-type), which has a wide range of biological activities and pharmacodynamic effects; meanwhile, the research value and high potential of patent medicine is predictable. Furthermore, P-b4, as a quality control marker for *Pulsatilla Chinensis* and documented by the Chinese Pharmacopoeia Commission (Chp, 2015, volume 1), has showed the highest content as a herb with pharmacological effects of antitumor [6], anti-inflammatory [7], antiviral [8], and immunomodulatory activities [9]. Nevertheless, currently, the anti-GA activity and metabolomic profiles on MSU crystal-induced rat model after being treated by P-b4 has not been reported.

Besides, as for GA, the complexity is constantly unraveled as with previous investigation, some metabolomics studies have proposed the pathogenesis, targets, and potential biomarkers have indicated that (1) IL-1β is the key mediator of gouty inflammation [10]; (2) the NLRP3 inflammasome is the major pathway for MSU crystal-induced inflammatory response [11]; (3) the release of IL-1β and TNF-α was decreased by the simultaneous downregulation of COX-2(Cyclooxygenase-2)/mPGES-1(microsomal prostaglandin E synthase-1), and 5-LOX (5-lipoxygenase) [12]; (4) AMP kinase (AMPK) is a pivotal regulator of processes that tend to mitigate against the GA related metabolic syndromes (Obesity, renal disease, cardiovascular disease) [13]. By extension, what is the mechanism of anti-GA activity of P-b4 and what is the biomarker between GA model group, dosing group and control group rats? These remain unknown and would intervene with the research of GA in clinical treatment. Therefore, a metabolomic profile on the effect of P-b4 on GA in an MSU crystal-induced rat model is urgently needed to identify the representative biomarker for the new drug R&D. P-b4 used in this study was extracted and purified form *Pulsatilla Chinensis* by our research group, as well as the purity reaching 98.50% (purity was calibrated by peak area normalization method). Meanwhile, the structures of P-b4 were identified by NMR (Avance 600, Bruker Daltonics Inc., Breca, MA, USA) and HRMS (SYNAPT-G2-S, Waters Corp., Milford, MA, USA). ^1^H-NMR, ^13^C-NMR, MS spectra, and the molecular structure of P-b4 is provided in Figure 2, and the detailed structure elucidation data (NMR) of p-b4 were shown in Appendix A.

At present, metabolomics have become a leading technology and effective tool for accurately identifying potential biomarkers of many diseases and different physiological and pathological statuses, leading to different biochemical states and metabolomic profiles. Meanwhile, comprehensive metabolite profiles, information on pathophysiological processes, pharmacological intervention, and metabolic kinetics could be obtained from complex biological samples by appropriate analytical method and multivariate analysis [14]. Unlike other omics, metabolomics reveal a collection of small endogenic molecules (molecular weight ≤1000 Da) [15] such as organic acids, amino acids, lipids, saccharides, and nucleosides, which tend to have biological effects at lower blood concentrations. Typically, non-targeted metabolomics studies were based on multitudinous high-throughput screening techniques. Taking ultra-performance liquid chromatography quadrupole time-of-flight mass spectrometry (UPLC-QTOF-MS/MS) for example, the advantages of this technology consist of high sensitivity/resolution and low ion suppression. Previous studies have provided massive accurate chemical information for thousands of compounds using UPLC-QTOF-MS/MS, which were a great benefit in the discovery of biomarkers and homologous pathways [16].

To investigate the detailed metabolomics profile of P-b4 ameliorates MSU crystal-induced GA rats model and to identify potential biomarkers, for the first time a serum metabolomics was applied using UPLC-QTOF-MS/MS with high resolution. In this study, we compared the serum metabolomics profiles on GA rats, GA rats ameliorated by P-b4, and healthy rats. In the meantime, a comparison of the anti-GA activity of P-b4 and positive drug (diclofenac sodium) was made in this study.

## 2. Results

### 2.1. The Major Pharmacodynamics Investigations (Anti-GA Ability) of P-b4

The 70 SD rats were grouped and treated as Animal Handling. Statistical analysis was performed using one-way analysis of variance (ANOVA). All tests were performed using SPSS 19 and a *p* value < 0.05 was considered statistically significant.

#### 2.1.1. The Effect of Each Group on Toes/Ankle Swelling

Compared with a control group, joint swelling in the GA model group was obvious 3 h after injection, peaked from 9 h to 12 h, and spontaneously alleviated at 24 h. Compared with the GA model group, the degree of joint swelling of rats in each P-b4 group decreased significantly from 9 h to 24 h. However, the degree of joint swelling of rats in diclofenac sodium group began to decrease significantly at 6 h, and the effect lasted to 24 h. 

#### 2.1.2. The Effect of Each Group on Pain Threshold

Compared with the control group, the pain threshold in the model group was significantly decreased; compared with the GA model group, the pain threshold of swollen feet in P-b4 groups increased in varying degrees with significant difference, while there was no significant difference between diclofenac sodium group and GA model group. The value of each group on toes/ankle swelling and pain threshold was provided in Appendix A and Table 1.

### 2.2. Validation of Analytical Method

Prior to the serum detection, the accuracy of the UPLC-QTOF-MS/MS system and the repeatability of method were validated by reduplicative analysis of 6 parallel injections of the same QC sample after 0/6/12/18/24/48 h. In addition, QC samples were analyzed once every 10 injections during serum sample detection in both the positive and negative modes. The total ion chromatograms (TICs) of QC samples showed almost complete overlap by visual inspection in both positive and negative mode (Appendix A). Besides, the responses (retention time, peak area, and intensity) of IS in the above 6 QC samples were stable within 48 h. Beyond that, blank solvent were injected once between two QC sample injections, the results showing that significant peak was non-detected in the blank solvent.

### 2.3. Changes in Serum Metabolites of the MSU Crystal-Induced GA Rats after Being Treated by P-b4 

#### 2.3.1. Multivariate Analysis of Rats Serum Metabolic Profile

Multivariate analysis was performed on SIMCA-P 16.1 with the processed data exported from PV. In order to figure out the intrinsic differences among the 7 groups (control, model, P-b4 groups (hypodermic, 20 mg × 4) at 9/24 h and QC group), the PCA and OPLS-DA were utilized to locate the distinguishing variables. The PCA score plots (Figure 3A,B) showed an unsatisfactory separating effect of data among the 7 groups both in the positive (R^2^X = 0.619, Q^2^ = 0.524) and negative (R^2^X = 0.648, Q^2^ = 0.508) ion modes. However, an important trend could be found after putting the 9 h and 24 h samples of each group into an elliptic region: the serum metabolic profile of MSU crystal-induced GA rats (model groups) was transforming to the normal SD rats (control groups) after being intervened by P-b4 (P-b4 groups).

The OPLS-DA score plots of all groups then (Figure 3C,D) showed a prominent separation both in the positive and negative ion modes, especially in the positive ion model. Moreover, the relevant R^2^Y and Q^2^Y values were applied to assess the quality of the OPLS-DA models. For rat serum metabolic profiles, R^2^Y and Q^2^Y were 0.964 and 0.851 in the positive ion mode and 0.927 and 0.788 in the negative ion mode. These values indicated that the presence of few irrelevant model terms and superior predictability parameters of the OPLS-DA models [17]. OPLS-DA was further performed to distinguish the obviously altered metabolites in two groups (12 pairs in total) as shown in Figure 4. The OPLS-DA scores plot showed clear separation between control and P-b4 groups/control and model groups/P-b4 and model groups at 9 h and 24 h in both positive and negative ion mode (Figure 4A–F). The R^2^ and Q^2^ of OPLS-DA models were provided in Appendix A, the CV-ANOVA of 12 OPLS-DA score plots were provided in Appendix A, and the permutation test was performed to validate OPLS-DA models (Appendix A).

#### 2.3.2. Identification of Potential Biomarkers for the Therapeutic Effect of p-b4 on MSU Crystal-Induced GA Rats 

Statistically distinct metabolites (VIP > 1, *p* < 0.05) were then screened for further discrimination using the online databases (see “2.6 Data Processing and Statistical Analysis”). The error between extraction mass value and experimental mass value was less than 5 ppm. The metabolites which were selected as potential biomarkers should conform to those prerequisites. The clustering heat maps (Figure 4A and Figure 5B) were performed for further understanding of the metabolic alterations in the rat serum samples after being intervened by P-b4 using Multi Experiment Viewer 4.9.0 (MeV) [18]. The identification of 32 potential biomarkers of rat serum samples among control groups, model groups, and P-b4 groups was provided in Table 2. Furthermore, arachidonic acid, 5(*S*)-Hydroperoxyeicosatetraenoic acid, sphingolipids (sphinganine, sphingosine, 3-dehydrosphinganine), glycerophospholipids (LysoPC(18:1(9Z)), phosphorylcholine, glycerophosphocholine), leukotrienes(leukotriene C4, and 12-Keto-tetrahydro-leukotriene B4 leukotriene A4) were identified as particularly significant biomarkers for the therapeutic effect of p-b4 on MSU crystal-induced GA rats.

#### 2.3.3. Analysis of Metabolic Pathways

Metabolic pathway analysis was contingent on all the 32 potential biomarkers using the KEGG and MetaboAnalyst 4.0 [19]. An overview of the pathway analysis was shown in Figure 5 and Figure 6 for rat serums which were sampled at 9 h and 24 h, respectively. The metabolic pathways of the rat serum samples were identified as arachidonic acid metabolism, sphingolipid metabolism, glycerophospholipid metabolism, valine, leucine, and isoleucine biosynthesis, aminoacyl-tRNA biosynthesis. The overview of metabolic pathway analysis was provided in Figure 7, and the pathway related data was provided in Appendix A.

## 3. Discussion

As a local inflammatory response, acute GA causes severe pain and swelling of the joints, leading to changes in endogenous serum metabolite levels. The clinical diagnosis of GA was generally based on joint swelling [20], CT (computed tomography) [21], and smear test, which was provided with uncertainties (MSU crystal smear of hyperuricemia was also presented positive results) and lack of clear diagnostic markers. On the other side, in clinical practice, anti-inflammatory drugs for acute GA mainly include corticosteroids [22], non-steroidal anti-inflammatory drugs (NSAIDs) [23], and colchicine [24], which are often accompanied by adverse reactions such as drug resistance, endogenous hormone inhibition, and irritation of the digestive tract, while having certain efficacy. As a result, it is imperative to accelerate the development of new drugs for acute GA and comprehensively understand the biomarkers for treating acute GA. In general, serum samples were widely used for discovery of potential biomarkers for types of arthritis such as rheumatoid arthritis (RA) [25,26], juvenile idiopathic arthritis (JIA) [27], psoriatic arthritis (PsA) [28], etc. Also, other types of biological samples were used in the study of biomarkers for arthritis, such as synovial fluid [29,30], urine [31], and tissues [32]. In this study, a serum metabolomic profiling strategy was adopted to illuminate the therapeutic mechanism of P-b4 in MSU crystal-induced GA rat model and identify potential biomarkers. From the perspective of rat foot swelling, pain threshold, and endogenous metabolites expression, this model was reliable and effectively manipulated, and could induce GA without impacting other tissues.

In this study, the main pharmacodynamics of different dosing methods and dosages of P-b4 was firstly investigated, the results showing that P-b4 possessed high anti-inflammatory activity in rats with GA induced by MSU crystals. Markerview 1.2.1 was used to export LC-MS original data, and PCA and OPLS-DA analysis were conducted after data normalization. The results have displayed that the OPLS-DA score plots of all groups (Figure 3C,D) showed a prominent separation in two ion modes, especially in the positive ion model. After further screening and comparison with HMDB, KEGG, and other online databases, 25 potential biomarkers were identified in the positive ion mode and 7 potential biomarkers were identified in the negative ion mode (Table 2). After the pathway enrichment of 32 potential biomarkers by MetaboAnalyst 4.0, it was concluded that p-b4 could be used to treat GA induced by MSU crystals in rats by regulating arachidonic acid metabolism, sphingolipid metabolism, glycerophospholipid metabolism, valine, leucine, and isoleucine biosynthesis, aminoacyl-tRNA biosynthesis (Figure 7).

Arachidonic acid metabolism was considered to be the most relevant metabolic pathway with P-b4 treatment effect in this study, and therein arachidonic acid (AA), 5(*S*)-hydroperoxyeicosatetraenoic acid (5(*S*)-HPETE), leukotriene C4 (L-C4), 12-keto-tetraene b4 (L-B4), and leukotriene A4 (L-A4) have been identified as particularly significant biomarkers. Studies have shown that AA can inhibit the inflammatory response caused by leukotriene, interleukin, prostaglandin E2, and histamine [33]. In this study, we found that no matter at 9 or 24 h, AA in GA group was significantly down-regulated compared with control groups (*p* < 0.001), and AA in P-b4 groups was significantly up-regulated compared with GA group(*p* < 0.005), meanwhile, 5(*S*)-HPETE, L-A4,L-B4 had the same trend with AA, L-C4 had the opposite trend with AA (Table 2). This suggested that P-b4 was able to treat MSU crystal-induced GA Rats by regulating AA in arachidonic acid metabolism.

In addition, sphingolipid metabolism and glycerophospholipid metabolism were closely bound up with the therapeutic effect of P-b4. The involvement of sphinganine, sphingosine,3-dehydrosphinganine, lysophosphatidylcholines (LysoPCs), phosphorylcholine (PC), glycerophosphocholine (GPC) was revealed through this study. Sphinganine could be used to inhibit the esterification of cholesterol induced by low-density lipoprotein (LDL) [34], which was obtained from 3-dehydrosphinganine through 3-dehydrosphinganine reductase (KDSR). LysoPCs was generated by the degradation of PC by lecithin cholesterol acyltransferase (LCAT) and could be degraded to GPC by lysophospholipase I (LYPLA1). The metabolic pathway networks of the above potential biomarkers were provided in Figure 8, and it could be seen that P-b4 had a significant regulatory effect on these metabolites, suggesting that the therapeutic effect of P-b4 was potentially closely related to sphingolipid metabolism and glycerophospholipid metabolism.

The metabolites of the rats serum sampled at 9 h were basically the same as those sampled at 24 h; meanwhile, a significant difference in metabolites between control groups and the GA groups was demonstrated by Figure 5 and Figure 6. Obviously, metabolite levels in the P-b4 groups were adjusted to be close to those in the control groups, and this phenomenon combined with the results of major pharmacodynamics could prove that p-b4 had a definite effect on MSU crystal-induced GA rats. However, the metabolites shown in Figure 5 were more diverse than those shown in Figure 6 and the detailed *p* value could be obtained from Table 2. This suggested that P-b4 needs to be administered continuously for more than 24 h.

From the perspective of the changes of metabolites in the GA group at 0–24 h, the effects of the GA model were consistent and steady. Nevertheless, the foot swelling and pain threshold data in Table 1 showed that the effects of GA model had subsided spontaneously at 24 h. It seems paradoxical, but it does reveal an important practical problem which is related to GA diagnostic criteria. After the apparent symptoms subside, the metabolites remain at the same level as during the GA attack, which may easily lead to the next GA attack or even a continuous one. Therefore, the regression of clinical symptoms or the decline of some biochemical indicators (such as blood uric acid) cannot be regarded as the end point of GA treatment. Actually, the retracement of biomarkers is the real sign of effectiveness. The 32 potential biomarkers of therapeutic effect of P-b4 in MSU crystal-induced GA rats proposed in this study can provide a certain basis for the clinical treatment and medication of GA. Furthermore, our research group will collect serum samples from GA patients with different disease stages for further metabolomics research, hoping to provide more guidance and suggestions for the development of P-b4.

## 4. Materials and Methods

### 4.1. Chemicals and Reagents

Acetonitrile for UPLC grade was purchased from Thermo Fisher Scientific (81 Wyman Street, Waltham, MA, USA). Deionized water was produced using a Mill-Q ultrapure water system (Millipore, Burlington, MA, USA). Formic acid for UPLC grade was obtained from Tianjin Kermel Chemical Reagent Company (Tianjin, China). Diclofenac sodium (Sustained release tablets) for positive control group was purchased from Novartis (Beijing, China). Monosodium urate for GA model group was purchased from Sigma-Aldrich CN (Shanghai, China). 2-Chloro-l-phenylalanine for using as an internal standard (IS) was provided by Macklin Biochemical Company (Shanghai, China). P-b4 was extracted form *Pulsatilla Chinensis* by our research group (Jiangxi University of Traditional Chinese Medicine, Nanchang, China), the purity reached 98.50%.

### 4.2. Monosodium Urate Crystal Preparation

5 mL of 1 mol/L sodium hydroxide solution and 800 mg of sodium urate were added to 155 mL of sterilized water for injection and boiled; the sodium urate was completely dissolved. The pH was adjusted to 7.0 with 1 mol/L hydrochloric acid, centrifuge immediately (3000 r/min, 2 min) after the solution appeared milky white, then the supernatant was centrifuged repeatedly until the crystals no longer precipitated out. The MSU crystals were collected and dried off at 60 °C (121 °C high pressure sterilization for 30 min before use), then dissolved in phosphate buffer saline (PBS,20mg/mL), the suspension can be seen as long spindle crystal under optical microscope (DM2500, Leica, Wetzlar, Hesse, Germany).

### 4.3. Animal Handling for MSU Crystal-Induced GA Rats Model

This research was accomplished in conformity to the guidelines of the experimental animal ethics committee (EAEC) of Jiangxi university of traditional Chinese medicine. The protocol was approved by the experimental animal ethics committee of Jiangxi university of traditional Chinese medicine. Specific pathogen free (SPF) Sprague–Dawley (SD) rats (female, 180–220 g) were obtained from the Hunan SJA Laboratory Animal Co., Ltd. (Changsha, Hunan, China). 

Previously, all rats were acclimated for 1 week under standard laboratory conditions. Afterwards, 70 rats were randomly divided into a GA model group, diclofenac sodium group (intragastric injection: 7.5 mg/kg), P-b4 group (hypodermic injection: 20, 10, 5 mg/kg), and control group, 10 rats in each group. The control group and the GA model group rats were treated identically with sterile physiological saline, and the dosing groups were given corresponding drugs according to weight (10 mL/kg). P-b4 was administered 1 h before modeling, once every 3 h, 4 times in total; Diclofenac sodium was dosed 1 h before modeling, once in total. Rats were injected with MSU crystals suspension (20 mg/mL) in the dorsal side of the right ankle at 1 h after the first dosing (0.2 mL/rat.). The control group was injected with sterile physiological saline (0.2 mL/rat.).

### 4.4. Rat Serum Sample Collection and Pretreatment

Eye orbital venous blood samples were collected into a 5 mL eppendorf tube (EP tube) from the rats (Control group, GA Model group and P-b4-20 mg × 4 group, 30 rats in total) at 9 h after establishing the GA model, respectively (The model was most effective), and 24 h after the last hypodermic. Serum was obtained after 20 min by centrifugation at 4500 r/min for 10min at 4 °C. All serum samples were stored at −80 °C and thawed before preparation.

A working IS solution of 2-chloro-l-phenylalanine (10.2 μg/mL) was prepared in methanol. 100 μL Serum samples were added to 400 μL of the working IS solution in 1.5 mL EP tube. A total of 1200 μL rat serum sample (20 μL of each rat serum sample) were added to 4800 μL of the working IS solution to generate a quality control (QC) sample for validating the reproducibility of the method and UPLC-QTOF-MS/MS stability [35]. Pretreated samples were vortexed for 3 min, placed on the ice for 30 min, and then centrifuged (15,000 r/min, 4 °C) for 10 min [36,37]. The supernatant was transferred into a sample bottle and stored at 4 °C for MS analysis of untargeted metabolomics.

### 4.5. LC-MS Conditions

The UPLC analysis was carried out on a Nexera X2 LC-30A instrument (Shimadzu Corp., Tokyo, Japan) equipped with an automatic degasser, a high pressure solvent delivery pump, diode array detector, column oven, and an auto-sampler. An ACQUITY UPLC^TM^ BEH C_18_ column (2.1 × 100 mm, 1.7 μm; Waters Corp., Milford, MA, USA) was applied for chromatographic separation. The mobile phases consisted of 0.1% formic acid/water (A) and acetonitrile (B). The mobile phase gradient was as follows: 95–80% A (0–3 min), 80–60% A (3–5 min), 60–40% A (5–9 min), 40–35% A (9–16 min), 35–20% A (16–18 min), 20–5% A (18–21 min), 5–95% A (21–23 min), 95% A (23–25 min). The flow rate was set to 0.35 mL/min, with a 5 μL aliquot of the analyte was injected, and the column oven set at 30 °C.

MS/MS detection was conducted on a Triple TOF^TM^ 5600+system (equipped with a Duo Spray source) for ions in both positive and negative modes (AB SCIEX, Foster City, CA, USA). In the positive mode, the electrospray ionization was applied with the following parameters: ion spray voltage, 4500 V; ion source temperature, 500 °C; Curtain gas, 25 psi; nebulizer gas (GS 1), 50 psi; heater gas (GS 2), 50 psi; and declustering potential (DP), 80 V. In the information dependent acquisition (IDA) experiment, the collision energy (CE) was set at 35 eV, and the collision energy spread (CES) was (±) 10 eV. In the negative mode, the electrospray ionization was applied with the following parameters: ion spray voltage, −4500V; ion source temperature, 500 °C; Curtain gas, 25 psi; GS 1, 50 psi; GS 2, 50 psi; and DP, −100V. In the IDA, CE was set at −30 eV, CES was (±) 10 eV. In both the positive and negative ion modes, the mass ranges were set at *m/z* 50–1250 Da for TOF-MS scans and TOF MS/MS scans. Dynamic background subtraction (DBS) was applied to match the IDA tests for UPLC-QTOF-MS/MS (AB SCIEX, Foster City, CA, USA).

### 4.6. Data Processing and Statistical Analysis

All serum samples were detected by UPLC-QTOFMS/MS and the raw data were preprocessed by MarkerView v1.2.1 (AB SCIEX, Foster City, CA, USA). The data preprocessing included deconvolution, peak normalization, and peak assignment. The retention time and intensity of each peak were determined using PeakView 1.2.0 with XIC manager (PV, AB SCIEX, Foster City, CA, USA). Then the processed data was imported into the SIMCA-P 16.1 (MKS Data Analytics Solutions, Sweden) for multivariate statistical data analysis. An overview was created by principal component analysis with DModX (PCA-X, unsupervised) at the first place, then orthogonal partial least squares discriminant analysis (OPLS-DA, supervised) was applied to distinguish the contribution degree of the detected variables between two groups (12 pairs in total). In addition, the one-way analysis of variance (one-way ANOVA) was further applied to test the significance of intergroup differentiations (*p* < 0.05) in OPLS-DA models, and a homogeneity test of variance (levene’s test) was performed prior to one-way ANOVA. After that, the metabolites were filtered by *p* value (*p* < 0.05) and Variable Importance in Projection (VIP > 1) which was calculated by Multi Experiment Viewer 4.9.0 (MeV) and SIMCA-P. The clustering heat maps were performed using MeV. Several online databases were applied to screen out potential biomarkers from the filtered metabolites afterwards, such as HMDB (http://www.hmdb.ca/), METLIN (https://isometlin.scripps.edu/), Metabolomics Workbench (http://www.metabolomicsworkbench.org), Chemspider (http://www.chemspider.com/). Pathway analysis was performed using MetaboAnalyst (http://www.metaboanalyst.ca/) and the Kyoto Encyclopedia of Genes and Genomes (KEGG; http://www.kegg.jp/). The network and phenotype based gene of GA was created and analyzed by phenolyzer (http://phenolyzer.wglab.org/) [38].

## 5. Conclusions

In this study, serum metabolomics profiling of the effects of P-b4 administration in an MSU crystal-induced GA rat model at both 9 h and 24 h was successfully established based on LC-MS as an appropriate technique for the identification of potential biomarkers. Moreover, the changes of metabolites and the self-extinction of model effects within 24 h revealed important information for GA diagnostic criteria: The regression of clinical symptoms or the decline of some biochemical indicators (such as blood uric acid) cannot be regarded as the end point of GA treatment. Our results suggest that the metabolic pathways of the therapeutic effect of P-b4 in MSU crystal-induced GA rats include arachidonic acid metabolism, sphingolipid metabolism, glycerophospholipid metabolism, valine, leucine, and isoleucine biosynthesis, aminoacyl-tRNA biosynthesis. Besides, AA, 5(*S*)-HPETE, L-A4, L-B4, L-C4, sphinganine, sphingosine, 3-dehydrosphinganine, LysoPCs, PC, and GPC were identified as particularly significant biomarkers. Furthermore, our research group plans to conduct further metabolomics research on the clinical course of GA.

## Figures and Tables

**Figure 1 molecules-24-03161-f001:**
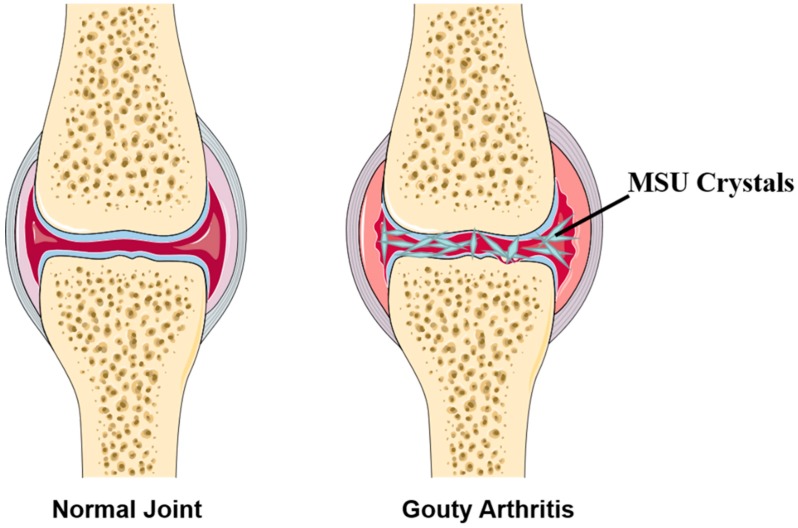
Abridged general view of normal joint and Gouty Arthritis (GA).

**Figure 2 molecules-24-03161-f002:**
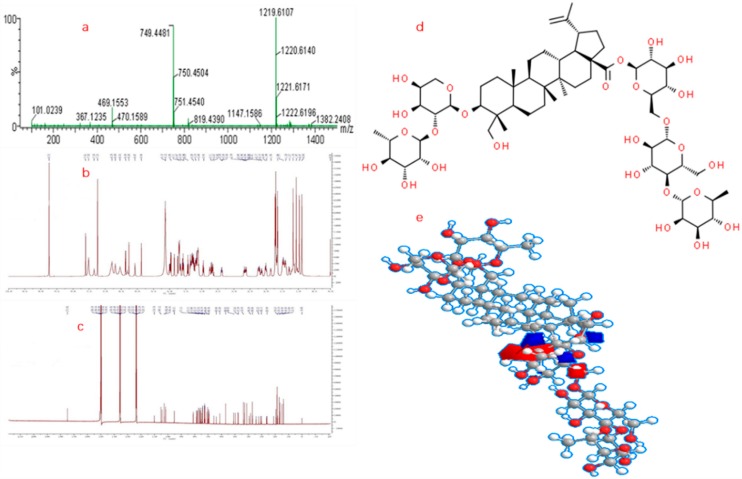
Molecular structure confirmation of P-b4: (**a**) MS spectra of P-b4; (**b**) ^1^H-NMR of P-b4; (**c**) ^13^C-NMR of P-b4; (**d**) 2D molecular structure of P-b4; (**e**) 3D molecular structure of P-b4.

**Figure 3 molecules-24-03161-f003:**
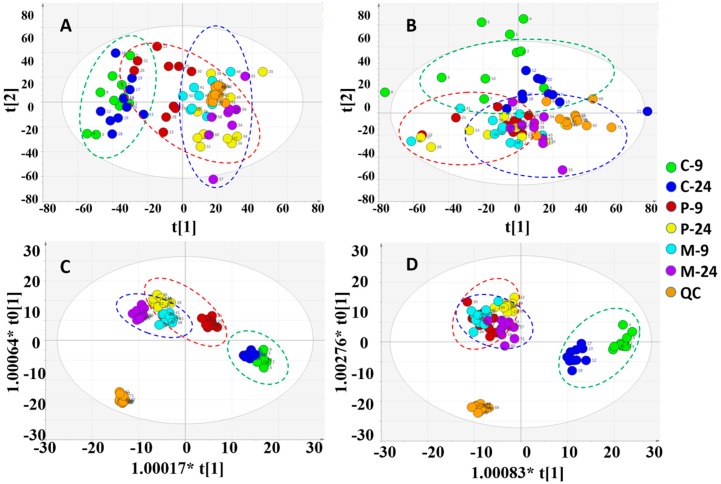
Overview of the rat serum samples: (**A**) PCA score plot in positive ion mode; (**B**) PCA score plot in negative ion mode; (**C**) OPLS-DA score plot in positive ion mode; (**D**) OPLS-DA score plot in negative ion mode (C-9: control group at 9 h;C-24: control group at 24 h; P-9: P-b4 group at 9 h; P-24: P-b4 group at 24 h; M-9: model group at 9 h; M-24:model group at 24 h; QC: quality control samples).

**Figure 4 molecules-24-03161-f004:**
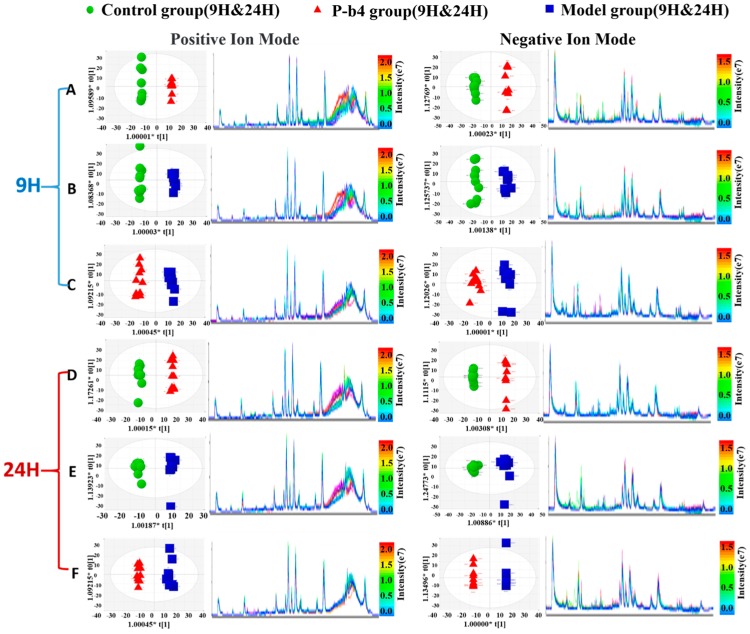
Orthogonal partial least squares-discriminant analysis score plots of samples (left panel) and TICs (right panel) obtained from different groups: (**A**) C-9 (green dots) VS P-9 (red dots); (**B**) C-9 (green dots) VS M-9 (bule dots); (**C**) P-9 (red dots) VS M-9 (bule dots); (**D**) C-24 (green dots) VS P-24 (red dots); (**E**) C-24 (green dots) VS M-24 (bule dots); (**F**) P-24 (red dots) VS M-24 (bule dots). The color bar on the right corresponds to the intensity of samples, beginning from weak (blue) to strong (red).

**Figure 5 molecules-24-03161-f005:**
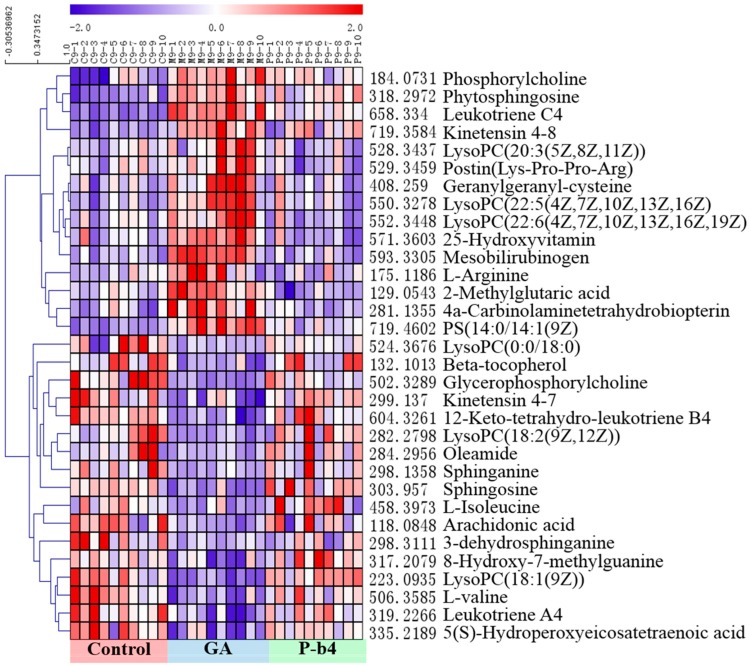
The heatmap of 32 potential biomarkers of rat serum samples at 9 h. The color of each section corresponds to a concentration value of each metabolite, red indicates an up-regulation of metabolites and blue indicates a down-regulation of metabolites.

**Figure 6 molecules-24-03161-f006:**
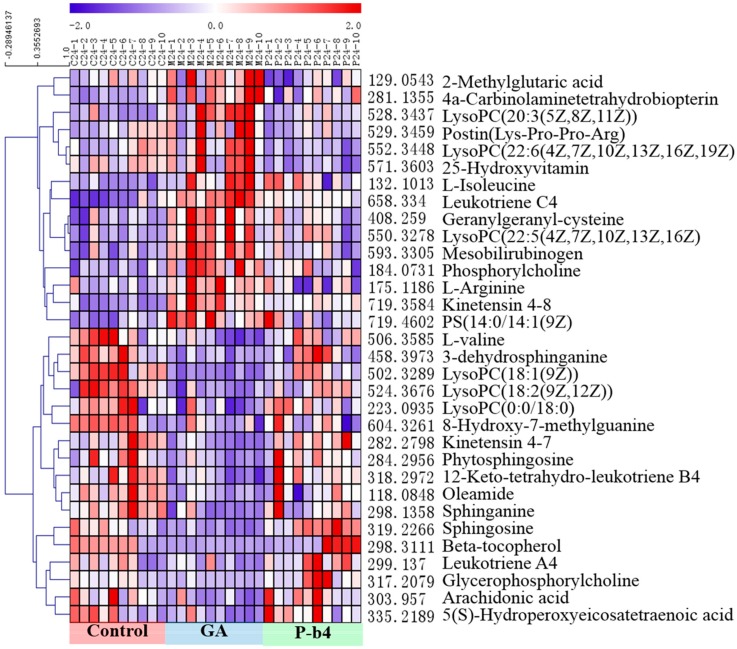
The heatmap of 32 potential biomarkers of rat serum samples at 24 h. The color of each section corresponds to a concentration value of each metabolite; red indicates an up-regulation of metabolites and blue indicates a down-regulation of metabolites.

**Figure 7 molecules-24-03161-f007:**
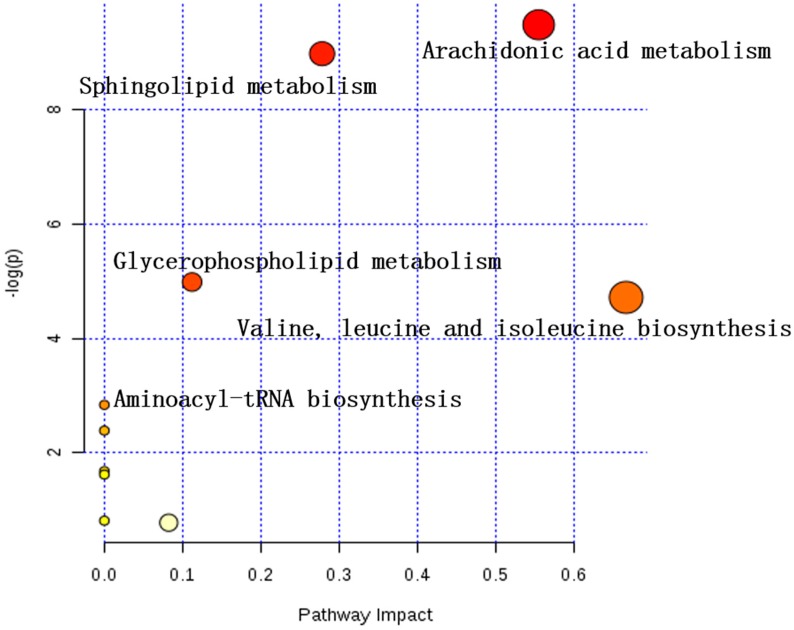
Overview of metabolic pathway analysis (MetaboAnalyst 4.0).

**Figure 8 molecules-24-03161-f008:**
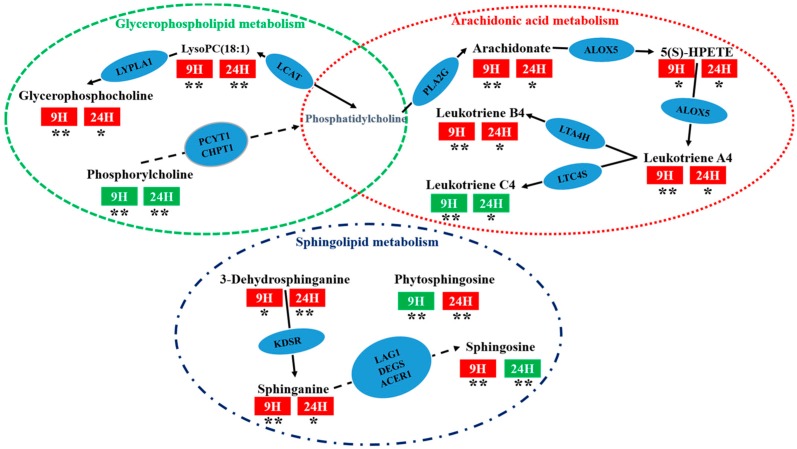
The metabolic pathway networks of particularly significant potential biomarkers for the therapeutic effect of p-b4 on MSU crystal-induced GA rats. The word in blue ovals means enzyme according to KEGG pathway database; the word in red and green rectangles means the sampling time of serum; red indicates an up-regulation of metabolites and green indicates a down-regulation of metabolites (GA model group VS P-b4 group); * means *p* < 0.05, ** means *p* < 0.01.

**Table 1 molecules-24-03161-t001:** Effect of each group on toes/ankle swelling and pain threshold in MSU crystal-induced GA rat model ($\bar{x}\pm s$, *n* = 10).

**Groups**	**Toes and Ankle Swelling (mL)**				
	**3 h**	**6 h**	**9 h**	**12 h**	**24 h**
Control	−0.13 ± 0.12 **	−0.16 ± 0.10 **	−0.16 ± 0.09 **	−0.16 ± 0.10 **	−0.18 ± 0.14 **
GA Model	0.16 ± 0.11	0.51 ± 0.22	0.95 ± 0.25	0.99 ± 0.22	0.47 ± 0.16
Diclofenac Sodium	0.10 ± 0.14	0.32 ± 0.14 *	0.51 ± 0.15 **	0.44 ± 0.17 **	0.08 ± 0.19 **
P-b4 (hypodermic, 20 mg × 4)	0.13 ± 0.11	0.44 ± 0.13	0.83 ± 0.20	0.84 ± 0.23	0.29 ± 0.15 *
P-b4 (hypodermic, 10 mg × 4)	0.11 ± 0.14	0.37 ± 0.18	0.69 ± 0.24 *	0.73 ± 0.27 *	0.46 ± 0.23
P-b4 (hypodermic, 5 mg × 4)	0.09 ± 0.12	0.41 ± 0.17	0.76 ± 0.29	0.79 ± 0.29 *	0.39 ± 0.12
P-b4 (intravenous, 1.2 5mg × 4)	0.11 ± 0.13	0.42 ± 0.17	0.68 ± 0.21 *	0.70 ± 0.14 **	0.27 ± 0.16 **
**Groups**	**Pain Threshold (s)**				
	**3 h**	**6 h**	**9 h**	**12 h**	**24 h**
Control	10.16 ± 3.05	9.00 ± 1.80 *	8.56 ± 1.49	8.08 ± 0.91 *	9.12 ± 2.27
GA Model	9.27 ± 2.25	7.17 ± 1.95	7.93 ± 1.95	6.86 ± 1.17	9.56 ± 1.32
Diclofenac Sodium	9.96 ± 1.58	7.52 ± 2.71	8.18 ± 1.63	8.11 ± 2.46	8.87 ± 1.64
P-b4 (hypodermic, 20 mg × 4)	10.10 ± 1.81	7.07 ± 1.09	6.80 ± 1.82	8.32 ± 2.44 *	10.27 ± 2.38
P-b4 (hypodermic, 10 mg × 4)	9.36 ± 1.90	8.22 ± 2.01	8.24 ± 2.35	8.08 ± 1.92 *	11.01 ± 2.44 *
P-b4 (hypodermic, 5 mg × 4)	11.57 ± 1.94 *	8.02 ± 2.62	8.70 ± 2.07	7.19 ± 1.51	9.63 ± 2.38
P-b4 (intravenous, 1.25 mg × 4)	10.73 ± 2.09	8.21 ± 1.75	10.13 ± 2.78 *	7.16 ± 0.97	9.81 ± 2.24

*p*-value is t-test comparing with GA model group rats; * means *p* < 0.05, ** means *p* < 0.01.

**Table 2 molecules-24-03161-t002:** Identification of potential biomarkers of rat serum samples among control groups, model groups and P-b4 groups.

NO.	Metabolites	RT	M.F	*p*-Value (9 h)	*p*-Value (24 h)	Trend			
						M-9 VS P-9	C-9 VS M-9	M-24 VS P-24	C-24 VS M-24
1	2-Methylglutaric acid	9.62	C_6_H_10_O_4_	1.95 × 10^−2^	3.43 × 10^−3^	↓	↑	↓	↑
2	l-Isoleucine	5.29	C_6_H_13_NO_2_	1.62 × 10^−3^	3.40 × 10^−3^	↑	↓	↓	↑
3	l-Arginine	11.90	C_6_H_14_N_4_O_2_	3.97 × 10^−3^	4.65 × 10^−3^	↓	↑	↓	↑
4	Phosphorylcholine	12.55	C_5_H_15_NO_4_P	2.38 × 10^−5^	3.27 × 10^−4^	↓	↑	↓	↑
5	8-Hydroxy-7-methylguanine	11.98	C_6_H_7_N_5_O_2_	1.95 × 10^−2^	2.88 × 10^−2^	↑	↓	↑	↓
6	4a-Carbinolaminetetrahydrobiopterin	9.62	C_9_H_13_N_5_O_3_	3.11 × 10^−4^	2.14 × 10^−3^	↓	↑	↓	↑
7	Oleamide	12.44	C_18_H_35_NO	1.18 × 10^−2^	2.18 × 10^−3^	↑	↓	↑	↓
8	Sphinganine	12.78	C_18_H_39_NO_2_	5.74 × 10^−3^	2.24 × 10^−2^	↑	↓	↑	↓
9	Glycerophosphorylcholine	22.46	C_8_H_20_NO_6_P	9.67 × 10^−3^	4.06 × 10^−2^	↑	↓	↑	↓
10	Phytosphingosine	9.73	C_18_H_39_NO_3_	1.79 × 10^−4^	3.26 × 10^−3^	↓	↑	↑	↓
11	Geranylgeranyl-cysteine	11.04	C_23_H_37_NO_3_S	3.96 × 10^−6^	5.79 × 10^−4^	↓	↑	↓	↑
12	Beta-tocopherol	20.28	C_28_H_48_O_2_	1.50 × 10^−2^	8.25 × 10^−3^	↑	↓	↑	↓
13	LysoPC(18:2(9*Z*,12*Z*))	11.43	C_26_H_50_NO_7_P	9.71 × 10^−12^	1.68 × 10^−7^	↑	↓	↑	↓
14	LysoPC(18:1(9*Z*))	16.79	C_26_H_52_NO_6_P	2.33 × 10^−5^	1.85 × 10^−3^	↑	↓	↑	↓
15	LysoPC(0:0/18:0)	16.81	C_26_H_46_NO_7_P	8.62 × 10^−4^	2.67 × 10^−4^	↑	↓	↑	↓
16	LysoPC(20:3(5*Z*,8*Z*,11*Z*))	16.79	C_28_H_50_NO_7_P	5.53 × 10^−8^	2.95 × 10^−4^	↓	↑	↓	↑
17	Postin(Lys-Pro-Pro-Arg)	12.42	C_22_H_40_N_8_O_5_	8.69 × 10^−4^	4.03 × 10^−2^	↓	↑	↓	↑
18	LysoPC(22:5(4*Z*,7*Z*,10*Z*,13*Z*,16*Z*)	11.98	C_30_H_52_NO_7_P	6.29 × 10^−6^	8.63 × 10^−4^	↓	↑	↓	↑
19	LysoPC(22:6(4*Z*,7*Z*,10*Z*,13*Z*,16*Z*,19*Z*)	11.34	C_30_H_50_NO_7_P	9.13 × 10^−7^	1.43 × 10^−3^	↓	↑	↓	↑
20	25-Hydroxyvitamin	11.98	C_34_H_52_O_8_	3.63 × 10^−5^	1.74 × 10^−3^	↓	↑	↓	↑
21	Mesobilirubinogen	11.98	C_33_H_44_N_4_O_6_	6.44 × 10^−7^	3.91 × 10^−7^	↓	↑	↓	↑
22	Kinetensin 4–7	4.82	C_26_H_37_N_9_O_6_	5.52 × 10^−3^	1.88 × 10^−5^	↑	↓	↑	↓
23	Leukotriene C4	4.82	C_30_H_47_N_3_O_9_S	2.05 × 10^−12^	3.87 × 10^−2^	↓	↑	↓	↑
24	Kinetensin 4–8	4.89	C_35_H_46_N_10_O_7_	2.05 × 10^−5^	4.71 × 10^−5^	↓	↑	↓	↑
25	PS(14:0/14:1(9*Z*)	20.85	C_34_H_64_NO_10_P	3.67 × 10^−9^	7.77 × 10^−4^	↓	↑	↓	↑
26	l-valine	9.94	C_5_H_11_NO_2_	3.49 × 10^−2^	1.15 × 10^−2^	↑	↓	↑	↓
27	Sphingosine	5.61	C_18_H_37_NO_2_	6.97 × 10^−3^	4.00 × 10^−5^	↑	↓	↓	↑
28	12-Keto-tetrahydro-leukotriene B4	12.93	C_20_H_34_O_4_	3.00 × 10^−4^	1.10 × 10^−2^	↑	↓	↑	↓
29	3-dehydrosphinganine	18.69	C_18_H_37_NO_2_	4.08 × 10^−2^	1.44 × 10^−3^	↑	↓	↑	↓
30	Arachidonic acid	19.99	C_20_H_32_O_2_	9.44 × 10^−4^	1.51 × 10^−2^	↑	↓	↑	↓
31	Leukotriene A4	13.40	C_20_H_30_O_3_	5.12 × 10^−3^	1.90 × 10^−2^	↑	↓	↑	↓
32	5(*S*)-Hydroperoxyeicosatetraenoic acid	10.93	C_20_H_32_O_4_	3.85 × 10^−2^	3.48 × 10^−2^	↑	↓	↑	↓

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
