# Peer review of "LC-MS Analysis of Serum for the Metabolomic Investigation of the Effects of Pulchinenoside b4 Administration in Monosodium Urate Crystal-Induced Gouty Arthritis Rat Model"

_molecules, 2019, doi:10.3390/molecules24173161_

Round 1

Reviewer 1 Report

The article of Shang Lyu and co-authors entitled "LC-MS Analysis of Serum for the Metabonomic Investigation of the Effects of Pulchinenoside b4 Administration in Monosodium Urate Crystal-Induced Gouty Arthritis Rat Model" reports a metabolic study of serum obtained from monosodium urate crystal-induced Gouty Arthritis rat model following Pulchinenoside b4 intervention. The overall study is very limited and the statistic analysis is weak. The number of replicates is very small considering authors are using OPLSDA. In figure 5, the pair wise analysis between treated and untreated mice should be reported using PCA instead of OPLSDA. Positive ion mode and negative ion mode should not be treated as 2 independent data sets but they should be combined as a single analysis. It is not clear how the list of the potential biomarkers have been validated; there are several uncommon biomarkers (e.g. kinetesin, postin, …). MS/MS fragmentation pattern and the acquisition of available standards should be included to confirm the identification of these metabolites, otherwise, a reader might be misled by the biomarkers reported in this study.

Also, the bibliography is very limited and additional articles relevant to arthritis and metabolomics field should be added including those that report the GA synovial fluid metabolomics analysis.

Author Response

Dear Reviewer :

  Thank you very much for your comments concerning our manuscript entitled “LC-MS Analysis of Serum for the Metabolomic Investigation of the Effects of Pulchinenoside b4 Administration in Monosodium Urate Crystal-Induced Gouty Arthritis Rat Model” (ID: 576447). Those comments are all valuable and very helpful for revising and improving our paper, as well as the important guiding significance to our researches. We have studied comments carefully and have made correction which we hope meet with approval. Revised portion are marked in red in the paper and supplementary material. The responds to the reviewer’s comments are as flowing:

Point 1: The number of replicates is very small considering authors are using OPLSDA. In figure 5, the pair wise analysis between treated and untreated mice should be reported using PCA instead of OPLSDA. Positive ion mode and negative ion mode should not be treated as 2 independent data sets but they should be combined as a single analysis. 

Response 1: An overview was created by PCA with DModX (unsupervised) at the first place, then OPLS-DA (supervised) was applied to distinguish the contribution degree of the detected variables between two groups (12 pairs in total). The R2 and Q2 of OPLS-DA models were provided in Table S2 and The permutation of 200 tests was performed to validate OPLS-DA models (Figure S3).

OPLS-DA was used to screen out compounds with VIP value greater than 1, which was more accurate for finding metabolites with significant difference. In addition, we believed that some metabolites responded well in positive ion mode, meanwhile other  metabolites responded well in negative ion mode. Therefore, we analyzed the positive ion model and the negative ion model respectively, so as to better understand the overall trend of metabolites.

Point 2: It is not clear how the list of the potential biomarkers have been validated; there are several uncommon biomarkers (e.g. kinetesin, postin, …). MS/MS fragmentation pattern and the acquisition of available standards should be included to confirm the identification of these metabolites, otherwise, a reader might be misled by the biomarkers reported in this study.

Response 2: In order to ensure the comprehensiveness and accuracy of the results, the retention time and intensity of each peak were determined using PeakView 1.2.0 with XIC manager (PV,AB SCIEX, Foster City, CA, USA), then the processed data was imported into the SIMCA-P 16.1 (MKS Data Analytics Solutions, Sweden) for multivariate statistical data analysis. After that, the metabolites were filtered by P value (P < 0.05) and Variable Importance in Projection (VIP>1) which was calculated by SIMCA-P. Several online databases were applied to screen out potential biomarkers from the filtered metabolites afterwards, such as HMDB (http://www.hmdb.ca/), METLIN (https://isometlin.scripps.edu/), Metabolomics Workbench (http://www.metabolomicsworkbench.org), Chemspider (http://www.chemspider.com/). Pathway analysis was performed using MetaboAnalyst (http://www.metaboanalyst.ca/) and the Kyoto Encyclopedia of Genes and Genomes (KEGG; http://www.kegg.jp/). The network GA was created and analyzed by phenolyzer (http://phenolyzer.wglab.org/).

  Furthermore, in order to convince the data, we added verification information of identified metabolites in the supplementary material (Table S4), which includes MS/MS fragmentation pattern, HMDB number and MS/MS matching score.

Point 3: Also, the bibliography is very limited and additional articles relevant to arthritis and metabolomics field should be added including those that report the GA synovial fluid metabolomics analysis.

Response 3: As for the lack of references, we have consulted many literatures related to arthritis and metabolomics, which have been added to the references of  this manuscript, meanwhile, the corresponding content was added in the discussion section.

  We tried our best to improve the manuscript and made some changes in the manuscript(in red).  These changes will not influence the content and framework of the paper.

  We appreciate for Editors/Reviewers’ warm work earnestly, and hope that the correction will meet with approval.

  Once again, thank you very much for your comments and suggestions.

Shang Lyu

                                                                                                   2019.8.16.

Reviewer 2 Report

The article entitled “LC-MS Analysis of Serum for the Metabonomic Investigation of the Effects of Pulchinenoside b4 Administration in Monosodium Urate Crystal-Induced Gouty Arthritis Rat Model” is evaluating the metabolic biomarkers of Gouty arthritis by mass spectrometry. The article needs extensive correction for English. Even in the abstract there several mistakes as given below.

The title has a spelling mistake. Metabolomic is spelled wrong. Pulchinenoside b4 need to be explained in the abstract. chromatographycoupled need to be chromatography coupled.

The introduction  and discussion needs to be better written for clarity

In the results section why the high concentration of p-b4 (20 mg*4) has higher swelling up to 12 h compared to p-b4 (10mg*4). What is the difference between the table and the graph? Both are showing the same data.

Explain why the PCA plot was not able to differentiate different groups.

Independent validation of the identified biomarkers is needed.

Author Response

Dear Reviewer :

  Thank you very much for your comments concerning our manuscript entitled “LC-MS Analysis of Serum for the Metabolomic Investigation of the Effects of Pulchinenoside b4 Administration in Monosodium Urate Crystal-Induced Gouty Arthritis Rat Model” (ID: 576447). Those comments are all valuable and very helpful for revising and improving our paper, as well as the important guiding significance to our researches. We have studied comments carefully and have made correction which we hope meet with approval. Revised portion are marked in red in the paper and supplementary material. The responds to the reviewer’s comments are as flowing:

Point 1: The title has a spelling mistake. Metabolomic is spelled wrong. Pulchinenoside b4 need to be explained in the abstract. chromatographycoupled need to be chromatography coupled.

The introduction  and discussion needs to be better written for clarity

Response 1: We are very sorry for our negligence of spelling mistakes in our manuscript, it should not have occurred, and I have corrected it immediately.

  The explanation of P-b4 has been added to the abstract, in addition, the introduction and discussion sections have been adjusted to enhance the organization.

Point 2: In the results section why the high concentration of p-b4 (20 mg*4) has higher swelling up to 12 h compared to p-b4 (10mg*4). What is the difference between the table and the graph? Both are showing the same data.

Response 2: Considering the Reviewer’s comments, We think the reasons for the high concentration of p-b4 has higher swelling up to 12 h compared to low concentration of p-b4 are as follows: (1) As a natural compound, the dose-response relationship of p-b4 is not clear and needs to be further studied. However, according to our repeated pharmacodynamic experiments, we speculate that the anti-inflammatory activity of p-b4 does not show a linear relationship with the dose;

(2) The trend reflected in the data in table 1 was consistent with our previous data in multiple trials, which could all show the significant anti-inflammatory activity of p-b4, which is also the purpose of setting " The Major Pharmacodynamics Investigations (Anti-GA Ability) of P-b4" in our manuscript;

(3) Meanwhile, the individual difference of experimental animals was also an unavoidable problem in pharmacodynamic experiments.

  It is really true as Reviewer suggested that Table 1 and Figure 3 were showing the same data, therefore, Figure 3 has been moved to the “supplementary material” and all figures have been renumbered.

Point 3: Explain why the PCA plot was not able to differentiate different groups.

Response 3: The main reasons for the unsatisfactory separation of different groups in the PCA score plots (Figure 3 A and B)  are as follows:(1) In order to intuitively reflect the changes of metabolites in different groups of 9h and 24h, we put the control groups, model groups and p-b4 groups of 9h and 24h into one PCA score plot.

(2)In fact, if the control groups, model groups and p-b4 groups of 9h and 24h were placed in two PCA score plots respectively, the separation of different groups was relatively ideal.

(3)However, an important trend could be found after putting the 9H and 24H samples of each group into an elliptic region: the serum metabolic profile of MSU crystal-induced GA rats(model groups) was transforming to the normal SD rats(control groups) after being intervened by P-b4(P-b4 groups).

Point 4: Independent validation of the identified biomarkers is needed.

Response 4: In order to ensure the comprehensiveness and accuracy of the results, the retention time and intensity of each peak were determined using PeakView 1.2.0 with XIC manager (PV,AB SCIEX, Foster City, CA, USA), then the processed data was imported into the SIMCA-P 16.1 (MKS Data Analytics Solutions, Sweden) for multivariate statistical data analysis. After that, the metabolites were filtered by P value (P < 0.05) and Variable Importance in Projection (VIP>1) which was calculated by SIMCA-P. Several online databases were applied to screen out potential biomarkers from the filtered metabolites afterwards, such as HMDB (http://www.hmdb.ca/), METLIN (https://isometlin.scripps.edu/), Metabolomics Workbench (http://www.metabolomicsworkbench.org), Chemspider (http://www.chemspider.com/). Pathway analysis was performed using MetaboAnalyst (http://www.metaboanalyst.ca/) and the Kyoto Encyclopedia of Genes and Genomes (KEGG; http://www.kegg.jp/). The network GA was created and analyzed by phenolyzer (http://phenolyzer.wglab.org/).

  Furthermore, in order to convince the data, we added verification information of identified metabolites in the supplementary material (Table S4), which includes MS/MS fragmentation pattern, HMDB number and MS/MS matching score.

  We tried our best to improve the manuscript and made some changes in the manuscript(in red).  These changes will not influence the content and framework of the paper.

  We appreciate for Editors/Reviewers’ warm work earnestly, and hope that the correction will meet with approval.

  Once again, thank you very much for your comments and suggestions.

Shang Lyu

                                                                                                   2019.8.16.

Reviewer 3 Report

in this work, non-targeted metabolomics analysis of  serum obtained from a MSU crystal-induced GA rat model thorough the antiinflamatory agent pulchinenoside b P-b4, using  ultra-performance liquid chromatography coupled to quadrupole time-of-flight tandem mass  spectrometry was performed.  the main pharmacodynamics of different dosing methods and dosages of P-b4 was firstly investigated, the results confirm that P-b4 possessed high anti-inflammatory  activity. These results demonstrated changes in serum metabolites with 32 potential biomarkers the work was done unsing multivariate analysys. the work is sound and parameters are ok. comments: please revise carefully the english language.

Author Response

Dear Reviewer :

  Thank you very much for your comments concerning our manuscript entitled “LC-MS Analysis of Serum for the Metabolomic Investigation of the Effects of Pulchinenoside b4 Administration in Monosodium Urate Crystal-Induced Gouty Arthritis Rat Model” (ID: 576447). Those comments are all valuable and very helpful for revising and improving our paper, as well as the important guiding significance to our researches. We have studied comments carefully and have made correction which we hope meet with approval. Revised portion are marked in red in the paper and supplementary material. The responds to the reviewer’s comments are as flowing:

Point 1: In this work, non-targeted metabolomics analysis of serum obtained from a MSU crystal-induced GA rat model thorough the anti-inflammatory agent P-b4, using ultra-performance liquid chromatography coupled to quadrupole time-of-flight tandem mass   spectrometry was performed.  The main pharmacodynamics of different dosing methods and dosages of P-b4 was firstly investigated, the results confirm that P-b4 possessed high anti-inflammatory activity. These results demonstrated changes in serum metabolites with 32 potential biomarkers the work was done using multivariate analysis. The work is sound and parameters are ok. comments: please revise carefully the English language.

Response 1: Thank you very much for your pithy summary and good comments on our manuscript, it is really true as you suggested that I should revise carefully the English language. I have improved the English of our manuscript and hope it can meet your requirements.

  We tried our best to improve the manuscript and made some changes in the manuscript(in red).  These changes will not influence the content and framework of the paper.

  We appreciate for Editors/Reviewers’ warm work earnestly, and hope that the correction will meet with approval.

  Once again, thank you very much for your comments and suggestions.

Shang Lyu

                                                                                                   2019.8.16.

Reviewer 4 Report

Manuscript ID: molecules-576447

The manuscript entitled: “LC-MS Analysis of Serum for the Metabonomic Investigation of the Effects of Pulchinenoside b4 Administration in Monosodium Urate Crystal-Induced Gouty Arthritis Rat Model”. Indeed, the current work could provide some positive scientific values, due to the basic information about serum metabolomics profiling of the effects of P-b4 administration in MS crystal-induced GA rat model. However, the manuscript contains many figures with too low resolution, which need to improve the sharpness of the images. On the other hand, the statistical analysis is few rigorous. The Tables are clear and sufficient providing the necessary information and the manuscript is well written.

Therefore, the manuscript could be accepted for publication after major revision. Please see specific comments below.

The authors need to improve the statistical analysis. The statistical analysis is few rigorous, it is not indicated which type of t-test was performed and if levene´s test was conducted prior to t-test to evaluate the homogeneity of variance of each metabolite.

All the figures in general have too low resolution.

In the line 116, table 1 showed effect of each group on toes/ankle swelling and pain threshold. The datas are normally distributed? the parametric statistical approach is the right choice? We don’t know anything about that.

Line 120, the figure 3 and table 1 showed the same information. I think that the figure 3 could be included in supplementary material.

Author Response

Dear Reviewer :

   Thank you very much for your comments concerning our manuscript entitled “LC-MS Analysis of Serum for the Metabolomic Investigation of the Effects of Pulchinenoside b4 Administration in Monosodium Urate Crystal-Induced Gouty Arthritis Rat Model” (ID: 576447). Those comments are all valuable and very helpful for revising and improving our paper, as well as the important guiding significance to our researches. We have studied comments carefully and have made correction which we hope meet with approval. Revised portion are marked in red in the paper and supplementary material. The responds to the reviewer’s comments are as flowing:

Point 1: The authors need to improve the statistical analysis. The statistical analysis is few rigorous, it is not indicated which type of t-test was performed and if levene´s test was conducted prior to t-test to evaluate the homogeneity of variance of each metabolite.

Response 1: The statistical analysis in our manuscript is One-Way ANOVA, and a homogeneity test of variance (levene´s test) was performed prior to One-Way ANOVA.

  The R2 and Q2 of OPLS-DA models were provided in Table S2 and The permutation of 200 tests was performed to validate OPLS-DA models (Figure S3).

  In addition, the CV-ANOVA of 12 OPLS-DA score plots were provided in supplementary material (Table S5).

   Furthermore, in order to convince the data, we added verification information of identified metabolites in the supplementary material (Table S4), which includes MS/MS fragmentation pattern, HMDB number and MS/MS matching score.

Point 2: All the figures in general have too low resolution.

Response 2: The resolution of all figures has been improved to above 450 PPI.

Point 3: In the line 116, table 1 showed effect of each group on toes/ankle swelling and pain threshold. The datas are normally distributed? the parametric statistical approach is the right choice? We don’t know anything about that.

Response 3: (1)The datas showed in Table 1 were normally distributed and were provided with homogeneity of variance.

  (2) Statistical analysis was performed using one-way analysis of variance (ANOVA). All tests were performed using SPSS 19 and a p value < 0.05 was considered as statistically significant.

Point 4: Line 120, the figure 3 and table 1 showed the same information. I think that the figure 3 could be included in supplementary material.

Response 4: It is really true as Reviewer suggested that Table 1 and Figure 3 were showing the same data, therefore, Figure 3 has been moved to the “supplementary material” and all figures have been renumbered.

  We tried our best to improve the manuscript and made some changes in the manuscript(in red).  These changes will not influence the content and framework of the paper.

  We appreciate for Editors/Reviewers’ warm work earnestly, and hope that the correction will meet with approval.

  Once again, thank you very much for your comments and suggestions.

Shang Lyu

                                                                                                   2019.8.16.

Round 2

Reviewer 1 Report

I have reviewed the revised version of the manuscript and appreciate all the efforts in trying to improve the manuscript. However, the statistics and the assignment remain a big issue for this manuscript. Also, I suggest the authors submit the paper to a journal of RA.

Reviewer 2 Report

N/A

Reviewer 4 Report

The manuscript could be accepted in present form.